# Stem Cell Biomarkers and Tumorigenesis in Gastric Cancer

**DOI:** 10.3390/jpm12060929

**Published:** 2022-06-03

**Authors:** Kenly Wuputra, Chia-Chen Ku, Jia-Bin Pan, Chung-Jung Liu, Yi-Chang Liu, Shigeo Saito, Kohsuke Kato, Ying-Chu Lin, Kung-Kai Kuo, Te-Fu Chan, Inn-Wen Chong, Chang-Shen Lin, Deng-Chyang Wu, Kazunari K. Yokoyama

**Affiliations:** 1Graduate Institute of Medicine, Kaohsiung Medical University, Kaohsiung 80708, Taiwan; kenlywu@hotmail.com (K.W.); r991046@gap.kmu.edu.tw (C.-C.K.); 1994220cfla@gmail.com (J.-B.P.); csl@kmu.edu.tw (C.-S.L.); 2Regenerative Medicine and Cell Therapy Research Center, Kaohsiung Medical University, Kaohsiung 80708, Taiwan; pinkporkkimo@yahoo.com.tw (C.-J.L.); kuoksfo@yahoo.com.tw (K.-K.K.); chwu@kmu.edu.tw (D.-C.W.); 3Cell Therapy and Research Center, Kaohsiung Medical University Hospital, Kaohsiung 80756, Taiwan; ycliu@kmu.edu.tw; 4Division of Gastroenterology, Department of Internal Medicine, Kaohsiung Medical University Hospital, Kaohsiung 80756, Taiwan; 5Department of Medicine, College of Medicine, Kaohsiung Medical University, Kaohsiung 80708, Taiwan; 6Saito Laboratory of Cell Technology, Yaita 329-2192, Japan; saict1@maple.ocn.ne.jp; 7Horus Co., Ltd., Nakano, Tokyo 164-0001, Japan; 8Department of Infection Biology, Graduate School of Comprehensive Human Sciences, The University of Tsukuba, Tsukuba 305-8577, Japan; kkato@md.tsukuba.ac.jp; 9School of Dentistry, Kaohsiung Medical University, Kaohsiung 80708, Taiwan; chulin@kmu.edu.tw; 10Division of General & Digestive Surgery, Department of Surgery, Kaohsiung Medical University Hospital, Kaohsiung 80756, Taiwan; 11Department of Obstetrics and Genecology, Kaohsiung Medical University Hospital, Kaohsiung 80756, Taiwan; tefu.chan@msa.hinet.net; 12Division of Pulmonary and Critical Care Medicine, Kaohsiung Medical University Hospital, Kaohsiung 80756, Taiwan; chong@cc.kmu.edu.tw

**Keywords:** biomarkers, cancer stem cells, gastric cancers, organoids, translational research

## Abstract

Stomach cancer has a high mortality, which is partially caused by an absence of suitable biomarkers to allow detection of the initiation stages of cancer progression. Thus, identification of critical biomarkers associated with gastric cancer (GC) is required to advance its clinical diagnoses and treatment. Recent studies using tracing models for lineage analysis of GC stem cells indicate that the cell fate decision of the gastric stem cells might be an important issue for stem cell plasticity. They include leucine-rich repeat-containing G-protein-coupled receptor 5 (Lgr5^+^), Cholecystokinin receptor 2 (Cckr2^+^), and axis inhibition protein 2 (Axin2^+^) as the stem cell markers in the antrum, Trefoil Factor 2 (TFF2^+^), Mist1^+^ stem cells, and Troy+ chief cells in the corpus. By contrast, Estrogen receptor 1 (eR1), Leucine-rich repeats and immunoglobulin-like domains 1 (Lrig1), SRY (sex determining region Y)-box 2 (Sox2), and B lymphoma Mo-MLV insertion region 1 homolog (Bmi1) are rich in both the antrum and corpus regions. These markers might help to identify the cell-lineage identity and analyze the plasticity of each stem cell population. Thus, identification of marker genes for the development of GC and its environment is critical for the clinical application of cancer stem cells in the prevention of stomach cancers.

## 1. Introduction

Gastric cancer (GC) is known for its high prevalence and poor prognosis among cancer deaths worldwide (see GLOBOCAN Cancer Fact Sheet [1]). The stomach epithelium is damaged by various environmental and endogenous stresses and is maintained by the dynamic homeostasis maintained by gastric stem cells. Damage of the epithelium region by an *Helicobacter pylori* infection, alcohol uptake, smoking, food, and other factors can lead to metaplasia such as spasmolytic polypeptide-expressing metaplasia (SPEM) and intestinal metaplasia (IM). Both metaplasia happens as alterations in the epithelium, which are induced by Trefoil factor 2 (Tff2) or Cdx2, as respective markers of the major initial stage of advanced GC [2,3]. The initial hits of SPEM and/or IM used to be accompanied with accumulated gene mutations in chronic gastritis, including vitamin deficiency, anemia, stomach ulcers, stomach lining erosion, and bleeding, and then derived to the diseases to severe adenocarcinoma [2,4]. To specify the diseases caused by IM, mucosal atrophy characterized by losing glandular elements resulted in replacing the connective tissue (fibrosis) or metaplastic cells (neoplastic lesion). These neoplastic lesions injured the gastric mucosa correlated with an *H. pylori* (infection) infection and resulted in chronic inflammation (chronic gastritis). Precancerous lesions also lead to SPEM, and are strongly correlated with GC progression [2].

The Cancer Genome Atlas (TCGA) has presented an understanding of gene mutation patterns in GC. However, many events related to disease chronology and the gene alterations and molecular commitments need to be elucidated [5].

In general, a biomarker is defined as “a biomolecule found in body fluids or tissues that is an indicative signature of a normal or abnormal process, or a disease, such as cancer.” Thus, monitoring of biomarkers is useful for tracing the progression from a normal condition to an abnormal state. Alterations in biomarkers can be due to various incidents, such as germline or somatic mutations, transcriptional changes, and epigenetic modifications. Many biomarkers have been identified and some are used as possible markers for cancer and disease [6]. Recently, a combination of the TCGA database and the gene expression omnibus (GEO) database was used to identify possible candidate biomarkers for cancer initiation and development, as well as drug resistance [7]. However, because of the heterogeneity of cancers, a series of candidates for each type of cancer stem cells has not been characterized during cancer progression.

We believe that overexpression of specific stemness genes in GCs may promote the self-renewal ability of CSCs in GC and is linked to patients’ prognosis. Several cell-surface markers such as CD133, CD44, CD166, and CD24, have been identified as gastric CSC markers [8]. Moreover, pluripotent factors such as OCT4, SOX2, and NANOG (KLF4 and c-MYC) and intracellular signaling pathways including WNT, sonic hedgehog (SHH), NF-kB, NOTCH, BMP, and TGFβ/SMAD factors are known as essential regulators of gastric CSCs [9]. Recently, the relationship between the overexpression of key CSC-related markers and possible signal axes with prognostic and clinical values in GCs has been recognized. However, there are some conflicting opinions on this relationship in GCs, because of the diversity in the methodology, sample sizes, and research participants. Thus, there is a need to better establish the relationship using meta-analysis and clinical data to generate new biomarkers [10].

In addition, knowledge of cell–cell interactions in morphogenesis and cancer development, and of stem cells and their stem cell niches, is important for understanding tumor formation and development [11].

In the case of GC, gastric organoid model systems are very useful for studying stem cells and their niches [12]. Organoids generated from the gastrointestinal (GI) system contain several mixed populations from epithelial tissues [13], mesenchymal tissues [14], and pluripotent stem cells (PSCs) [15]. Such culture systems of GI organoids are possible for generating 3D structure-like gland crypts and microenvironments by usage of defined factors, such as defined niche factors, Wnt-R-spondin, epidermal growth factor (Egf), and Noggin [13,14,15,16,17,18]. These factors activated the niches of the GI epithelium to produce PSCs and epithelial surrounding cells in vitro. Thus, the creation of GI 3D models has merit to allow the simulation of the stomach microenvironment in vivo; in this regard, the critical biomarkers are important to define cell plasticity [19,20].

In this review, we describe the roles of gastric cancer stem cells (GCSCs) and their stem cell niches in GCs, and how to alter their genetic traits, as well as their potential application in preclinical therapies of stomach cancers. An understanding of these specific biomarkers in GCSCs and the stem cell niches is critical for studying cell plasticity during GC development.

## 2. Potential Biomarkers of Stem Cells in the Sub-Regions of the Stomach

The human stomach is made up of anatomical sub-regions: the antrum, corpus, cardia, and fundus (Figure 1). These tissues are made from foregut endoderm and have a critical function for digestion by the initial secretion of acids and enzymes. In the developmental steps, the cell communications between mesenchymal cells and epithelial cells are important to specify the cell types and tissue patterning through the cell proliferation, differentiation, and cell death in the indicated area mediated by the specific transcription factors, respectively. Respective functions of stem cells can be clarified by usages of human biopsy specimens and recombinant mice [21]. The stem cells’ progenitors produce the progenies which lead to replicating toward the apical and basal glands bi-directionally. Thus, we think that the pluripotent progenitor cells are initially localized in the isthmic regions of glands, which are localized in the antrum and corpus [22,23] (Figure 1). Both isthmic areas of glands contain the immature cells with indicated granules which function as the lineage-committed progenitor cells.

In the corpus gland in mice, undifferentiated stem cells are rare, and granule-free cells exist in the gland’s isthmus [24]. In addition, similar stem cells are detected in the isthmic locus in the antral glands [23,25].

Thus, the immature GSCs, which showed the different types of granules, act as independent lineages in both the antrum and corpus regions.

### 2.1. Antrum Stem Cells

#### 2.1.1. Lgr5

Lgr5 is detected in epithelial stem cells in several tissues including the stomach [2,26]. Lgr5^+^ epithelial stem cells provide a source of cancers in the epithelium, colon, and stomach, and tumor-resident Lgr5^+^ cells in intestinal cancers can be functional stem cells responsible for the progression of cancer and related diseases. Tumor-resident Lgr5^+^ cells exist in GC, but their cellular and genetical features as CSCs have not been established. Analysis of Lgr5^+^ CreERT recombinant Cre mice revealed that Lgr5^+^ stem cells can be found at the basement of the antral gland, especially in the basal crypt regions +1 to +3, which are localized as proliferation-active regions below the antral isthmus. They function as the basal stem cells [12,27]. In addition, Lgr5^+^ subpopulations reside within the isthmus [28].

#### 2.1.2. +4 Stem Cells (Cckbr, Mist1, Bmi1, eR1)

In the isthmus area of the antrum, several +4 antral stem cells have been detected and are specified by biomarkers: cholecystokinin B receptor (Cckbr), muscle intestine and stomach expression 1 (Mist1), Bmi1, and estrogen receptor 1 (eR1). In general, these stem cells can proliferate more rapidly than antral Lgr5^+^ stem cells [27,28,29,30,31].

#### 2.1.3. Axin2^+^ (a Wnt Target) Stem Cells

These cells are divided into (a) Lgr5^+^ basal region stem cells and (b) Lgr5^+^ and +4 isthmus region stem cells which are targeted by R-spondin. In Axin inhibition protein 2^+^(Axin2^+^)-CreERT mice, Axin2^+^ and CXCR4^+^ chemokine receptor (CD184^+^) cells occupy the lower third of the antrum, which contained both regions of the isthmus and basal [27,29]. The group of Axin2^+^ consists in different stem cells such as the Lgr5^+^ basal region and Lgr5^−^ +4 isthmus region. The Axin2^+^Lgr5^−^ isthmus region stem cells are mainly R-spondin targets [27]. Most Lgr5^+^ stem cells can proliferate and divide symmetrically into two stem cells. Thus, the Lgr5^+^ stem cells are kept by neutral drift mechanisms with competition force.

#### 2.1.4. Cckbr^+^ Stem Cells

One type of +4 stem cells, Cckbr^+^ stem cells, is committed to the asymmetric division to generate the progeny with canonical notch ligand 1 and either Numb or Delta for long- term proliferation and division [32]. By contrast, in the cases of inflammation and carcinogenesis, the Cckbr^+^ cells can be converted to the symmetric cell division to expand the stem cell pool for tissue regeneration, probably in part by the loss of gastrin-dependent signals [32]. Gastrin secreted from G cells resided near Cckbr+ stem cells which expressed gastrin receptor. In the case of the loss of gastrin, Cckbr+ stem cells became more proliferate and started to divide symmetrically, indicating that gastrin kept maintaining quiescence or inhibited antrum stem cells.

### 2.2. Corpus Stem Cells

Mist1-CreERT2 mice were used to identify corpus granule-free stem cells [33]. Mist1 (Bhlha 15) is known as a marker of gastric chief cells but has been shown using in situ hybridization studies to identify the rare solitary cell within the corpus isthmus. Mist1+ gastric isthmus stem cells in the corpus were regulated by the Wnt5a-dependent stem cell niche in response to injury and ulceration of the stomach [34].

#### 2.2.1. Mist1 (bhlha15)

Mist1 is a biomarker of chief cells in the stomach. Mist1^+^ isthmus stem cells have a slow proliferation rate and a lower cell division rate [33]. They can generate daughter cells from the isthmus in both directions and then spread into the entire corpus gland region. However, they divide and proliferate slowly and are relatively quiescent. In addition, they were also recognized as zymogenic or chief cells. Investigations are further required to characterize Mist1^+^ isthmus cells in more detail to clarify the slow division.

#### 2.2.2. Isthmic Stmn1/Iqgap3-Positive Cells

Isthmus corpus stem cells with markers such as Bmi1, eR1, Stathmin1 (Stmn1), and Ras GTPase-activating-like protein-IQ Motif Containing GTPase Activating Protein (Iqgap) [28,30,31,35,36,37] can proliferate well and are recognized as Ki67^+^ positive cells. They can produce all lineage of cells developmentally in the corpus including chief cells, mucous neck cells, parietal cells, and pit cells [37].

Stmn1^+^ cells existed in the heterogeneous cells. Bmi1^+^ cells existed in the quiescent cells. If epithelial cell injury occurred, they entered the cell cycle actively to proliferate rapidly. In a similar way, isthmus corpus cells, either Mist1, eR1 or Stmn1, entered to cell division rapidly if they had acute damage or *kRas* gene mutations [30,31,36,37].

#### 2.2.3. Trefoil Factor 2 and Cckb-Receptor Positive Cells (Tff2^+^ and Cckbr^+^)

These cells are known as the short-lived and lineage-committed stem cells. Cckbr was detected in both Tff2^+^ cells and parietal cells, which can secrete acids, as well as in enterochromaffin-like (ECL) cells that can release histamine. In the response to gastrin, they can mediate acid-secretory functions. However, such stem cells were also detected in the corpus isthmus, and the cell-growth ability of Cckbr^+^ isthmus stem cells was enhanced during hypergastrinemia [38]. Under normal conditions, Cckbr^+^ isthmus cells are known as the cell origin of ECL cells that accumulate and propagate during chronic hypergastrinemia [38]. Thus, Cckbr^+^ cells in the corpus regions are functionally distinct from those in the antrum.

#### 2.2.4. Claudin18 (Claudin18-IRES-CreERT2)

This Claudin18-IRES-CreERT2 driver mouse provides conditional mutations selectively to both pyloric and corpus regions in the gastric epithelia [18]. Fatehullah et al. [39] generated a *Claudin18-IRES-CreERT2* allele to drive conditional expression of the Wnt, RTK, and Trp53 axis in the stomach epithelium. In parallel, they produced the orthotopic cancer organoid transplantation models to evaluate tumor-resident Lgr5^+^ stem cells via in vivo ablation. Thus, Claudin18 tumors recapitulate human advanced gastric cancer precisely in terms of disease morphology, aberrant gene expression, markers, and sites of distant metastases. The tumor-resident Lgr5^+^ stem cells are critical to the initiation and maintenance of tumor and are obligatory for establishing the metastasis in GC in vivo.

## 3. Stem Cell Niches

### 3.1. Antral Stem Cell Niches

Interaction between antral GSCs and their stem cell niches are described below (Figure 2 and Figure 3).

R-spondin3 is secreted from myofibroblasts and acts as the niches of antral stem cells through the activated Axin2^+^ +4 stem cells [27,28].Antral Lgr5^+^ stem cells in the basement, or aquaporin-5 (Aqp5)^+^ basal stem cells are also stimulated by Wnt signals to associate with the Frizzled-7 receptor [40,41].G cells and Cckbr^+^ stem cells, which have gastrin receptors near G cells, can secrete the gastrin. Gastrin deficiency causes the Cckbr^+^ stem cells to proliferate extensively and to undergo division symmetrically [42] suggesting that gastrin inhibits antral stem cells and/or maintains them in a quiescent state [42].Enhancement of the Notch signal in the antral glands causes the deletion of G cells and increases the cell growth of Cckbr^+^ stem cells, which is gastrin. Knockout of the mouse gastrin gene stimulates oncogenesis in the antrum, and then, the antral G cells can provide the critical niches to inhibit the expansion of GSCs for GC progression [42,43].Antral Lgr5^+^ stem cells express cholinergic neuron-dependent muscarinic acetylcholine receptors, and then tuft cells indeed produce choline acetyltransferase [43].Tuft cells also secrete prostaglandins [44,45] from stromal fibroblasts and then enhance the stemness of GSCs as well as intestine SCs [46].Both basal- and isthmic-antral stem cells have the CXCR4 receptor, where their ligand CXCL12 is produced from endothelial vascular cells near GSCs [30].The stem cell niches and microenvironments can be proliferated and propagated during tumorigenesis, which promote stem cell division symmetrically [33].

### 3.2. Stem Cell Niches in Corpus

The corpus stem cells are also controlled by niches and their microenvironments.
Four growth factors, Egf, Noggin, R-spondin, and Wnt are essential for the long-term cultivation of 3D organoids from the gastric corpus [18,45], indicating that they might exist in the niches of this corpus, as in other regions of the GI tract (Figure 3).Notch in the niches of corpus stem cells is critical because activated Notch can promote the proliferation of the isthmic cells and their tumorigenesis [46].The EGF receptor (EGFR) and its ligands, and *Ras*/*Raf* mutations, both lead to robust proliferation of isthmus cells [47,48,49,50]. If the expression of EGFR is restricted to the upper third of the corpus glands [38], EGFR is a key player that controls the function of the corpus isthmus stem cells, although the location of cells producing EGFR ligands is not known.The Noggin/bone morphogenetic protein (BMP) pathway might act to stimulate corpus stem cells because the unique Gremlin1-expressing stromal cells reside near the corpus isthmus stem cells [51].Target molecules of Wnt/R-spondin such as Lgr5 and Axin2 are dominantly expressed in the basement region of the corpus gland because the stromal cells nearby produce R-spondin [28,52].The noncanonical Wnt5a ligand supports corpus stem cell activation, although there is a strong dependence on the canonical Wnt. Noncanonical Wnt5-positive type 2 innate lymphoid cells (ILC2s) were present near the stem cell area and activated isthmus stem cells [53]. In regenerative conditions and inflammatory states, Wnt5a^+^ ILC2a cells accumulate near the stem cell zone, in response to interleukin-33 (IL-33) and CxCL12 produced from surface pit cells and endothelial cells, respectively, which contribute to activation of isthmus stem cells [33,54]. Frizzled-5, a putative Wnt5a receptor, exists in isthmus stem/progenitor cells, indicating that Wnt5a may play a role in activating stem cells via this receptor [34]. Further study of Wnt5a is required to demonstrate modulation of corpus progenitors. The Wnt/R-spondin axis performs the essential role in niche function in corpus stem cells.

## 4. Gastric Chief Cells Function to Serve as Potential Reserves

The corpus glands contain different cell types that display a similar function to stem cells, as observed in the intestine, particularly when the glands have been damaged, and the function of normal stem cells is diminished. Gastric chief cells act as a reservoir of corpus stem cells after injury and cell damage (Figure 2) [38]. The abilities of dedifferentiation and limitation of DNA synthesis of gastric chief cells has been presented to happen as the “base stem cells” [55]. Chief cells express several marker indexes such as Lgr5, Mist1, and Troy, and the specific marker Gpr30 [26,53,56,57]. However, several points are described to explain the transition of chief cells into stem cells. Further investigation is required. These issues include the following: (i) although recombinant mice such as Mist1-CreERT and/or Lgr5-CreERT display chief cells, the transgene is expressed in some cell populations in the isthmus. Previous reports showed the lineage tracing occurred in the isthmus rather than chief cells [58]; (ii) exposure to high concentrations of tamoxifen with CreERT results in a toxic event and produces off-target events, decreasing the specific labeling of cells and complicating the results of the corpus lineage events [26]; and (iii) in most gastric injury models, it is critical to test the actions on stem cells in the isthmus and the cell growth within the niches following injury [38,58,59]. Therefore, the early inspection on chief cells and isthmus stem cells is necessary to be examined in the gastric injury model.

In the case of the gastric injury models with high-concentrations of tamoxifen, chief cells demonstrated abnormal kinetics, being depleted from acute injury rapidly. Thus, to study the cell-fate of chief cells after injury, it is necessary to perform genetic tracing, which is independent of tamoxifen, and for this purpose, a tetracycline-inducible (reverse tetracycline transactivator (rTA) and tet-ON) protocol was demonstrated. A model of two different progenitor regions similar to the isthmus and basal layers has been described in the proximal fundus stomach [33], and a long-term cell lineage-tracing study suggests that a subpopulation of chief cells may have features of long-term survival and self-renewal to produce clones which are not dependent of the isthmus-derived lineage [58,59].

## 5. Cellular Origins of Antral Gastric Cancers

The mutation of adenomatous polyposis coli (*Apc*) in stem cells resulted in adenomas or intramucosal dysplasia [13,30,60]. Multiple alterations of oncogenic drivers, including *Apc*, *p53*, *Kras*, *Pten*, or *Smad* genes, were reported to produce the invasive tumors derived from antral stem cells [32,41,61]. Thus, driver mutations in the antral stem cells are essential for initiating gastric carcinogenesis (Table 1). In general, these stem cell markers are critical for the identification of the gastric stem cells in different regions of the stomach, although some of them are the markers for the epithelial cells, such as Lgr5 [62]. In these cell types, the driver mutations happen to generate these normal stem cells to the cancer stem cells to enter the neoplastic stages, which are shared with other types of cancers.

In the antrum, Lgr5^+^ basal stem cells and/or +4 stem cells can produce oncogenic mutations for cancer initiation in mice. An *H. pylori* infection rapidly changes the antrum stem cell niches to produce R-spondin 3 from myofibroblasts, which causes the cell division of Axin2^+^ +4 stem cells in the antrum region symmetrically [28]. The Lgr5^+^ stem cells also committed to expand later in response to R-spondin [62,63], which also secreted to promote differentiation [52]. Thus, both antral stem cells and their niches are changed, and they can expand clonally after exposure of carcinogenic ligands for GCs. In tuft cells, acetylcholine-dependent nerve signaling is generated during tumorigenesis and a cholinergic innervation resulted in induction of the clonal propagation of GSCs-derived clones. Moreover, the injury activated the Notch cascade by exposure of N-methyl-N-nitrourea in the antrum region to decrease the number of gastrin-producing G cells and, by contrast, increase the number of Cckbr^+^ stem cells [32].

## 6. Oncogenic Transformation of Isthmic Stem Cells in Corpus

Corpus isthmic stem cells are reported to be less sensitive to oncogenic transformation than antrum stem cells [33]. Single DNA mutations in the corpus stem cells are not sufficient to produce dysplasia or cancers. For example, Anoikic Mist1 stem cells committed apoptosis following loss of the *Cdh1* gene, and then the cell number of *Cdh1*-deficient Mist1^+^-stem cells was gradually reduced. However, in the case of inflammation, which was induced by *H. pylori*, *Cdh1*-deficient Mist1-derived stem cells were able to proliferate and expand, and then finally cause gastric cancers, in a similar case of human signet ring cell GCs [33]. In part, this event can be explained by the increased expression of Cxcl12 and division of ILC2 cells via Wnt5a.

Additional mutations of *p53* or *Rhoa* genes in *Cdh1*-deficient Mist1^+^ -stem cells accelerated to form diffuse-typed cancers [33]. Although the *Apc* mutation alone in corpus stem cells cannot produce the tumorigenesis at this site, the mutations in both *Kras* and *Apc* genes at the same time caused GCs production efficiently. Thus, an *H. pylori* infection and *Kras* mutation produced gastric atrophy, leading to the disappearance of both parietal and chief cells and then subsequently generated metaplasia. Such abnormalities are required for cancers derived from isthmic stem cells in the corpus region (Table 1). It is reported that the deletion of parietal cells increased the expansion of the isthmus-derived stem cells to generate the gastric atrophy, which preceded tumorigenesis in humans [32]. Therefore, we expect that stem cells from the glandular isthmus act as the major source of GCs after interacting with other microenvironments.

## 7. Gastric Stem Cells and Their Niches

In general, a specific niche-microenvironment is the key factor to control the action of stem cells. The niches are composed in stem, stroma, immune cells, and extracellular matrices, which include various growth factors (Figure 3). Thus, it is complicated to define the culture condition in vitro. In gastric development, two of the major regulators are the sonic hedgehog (Shh) and BMP signaling pathways. In the human stomach, expression of Shh is limited to parietal cells and the glandular portion [64], where it controls the proliferation of the epithelial parts and produces a polarizing signal for differentiation of fundic glands via autocrine mechanisms in part [65]. BMPs are mainly produced from interstitial myofibroblast-like cells [66]. Several studies have reported that abnormalities of the BMP pathway in the stomach can cause perturbation of normal homeostasis around gastric mucosa, resulting in the development of metastasis, dysplasia, and neoplasia [66].

Notch signaling is also critical in this surrounding effect [46,67]. In general, Lgr5^+^ antral stem cells are regulated by Notch 1 and 2 receptors, but this signaling is reduced in the differentiation of the antral gland [41,68]. In the gastric antrum, Notch influenced the dynamic balance between cell growth and differentiation of stem cells and maintains the homeostasis of the gastric epithelium circumstances. Thus, alteration of Notch signaling can enhance the transformation activity of gastric normal stem cells to cancer stem cells [68].

In addition, growth factors such as Egf, fibroblast growth factor 10 (Fgf10), and Wnt also play a critical role in the control of self-renewal of GSCs [69]. Wnt signaling is required for keeping the undifferentiated phase of progenitor cells in the isthmus corpus, but its activating signal propagates the progenitor cells in the antrum [70]. Thus, the antral Lgr5^+^ stem cells are partially sustained by Wnt through frizzled-7 receptor [40] and, in addition, the corpus Mist1^+^ isthmus stem cells can be partially stimulated via the Wnt5a pathway [34]. Thus, Wnt activation might be closely related with gastric carcinogenesis in mouse gastric stem cells [70,71].

The mesenchymal effect in the glands has been less well-studied. The immune or stromal cells present in the gastric epithelium are included as the additional environmental effects for gastric stem cells. Both Cxcl2^+^ endothelial cells and Cxcr4^+^ innate lymphoid play roles in the niches of the corpus stem cells, in part via secretion of Wnt5a from innate lymphoid cells [34]. R-spondin 3 is produced from myofibroblasts and is a critical factor of the antral stem cell niche, and thus mainly activates Axin2^+^/Lgr5^−^ stem cells [28]. Gastrointestinal hormones, including gastrin and acetylcholine, may affect the function of the antral stem cell niche. Gastrin is produced from G cells near the antral isthmus, and Cck2R is the receptor for gastrin and progastrin under normal status. Cck2r^+^ stem cells have been identified in this area, and progastrin, but not gastrin, can enhance the growth of Cck2r-expressing stem cells [72]. Acetylcholine generated by tuft cells in the gastric epithelium controls the DNA synthesis, and regeneration of the gastric epithelial cells, as well as clonal expansion of Lgr5^+^ stem cells through the muscarinic receptor subtype 2 (M2R) [45,63].

Although we understand the essential roles of these growth factors and cell components in gastric stem cell niches, we are still uncertain when identifying which cells provide these niche factors and how they act with each other. These questions remain elusive. Furthermore, information of the microenvironments near the basement of the corpus gland is also limited. Thus, further studies are required to understand the components involved in the gastric stem cell niches and GSCs, and characterize their roles in the controlled homeostasis of the gastric epithelium.

## 8. Guidelines for Standardizing Novel Organoid Systems

Several procedures are generated to develop epithelial organoids for the same cell source, and standardization is now required. Some guidelines for standardization include the following: (i) the reproducible culture conditions should be defined, recombinant medium and the recombinant growth factors and small molecules should be used, and fluorescence-activated cell sorting should be used to purify the indicated cell sources; (ii) the organoid systems should be clarified gene expressions and their actions in long-term culture; and (iii) a high-quality dataset such as single-cell data of organoid primary tissue and their corresponding organoids should be constructed. For human translation, the Organoid Cell Atlas [73], which was recently launched as a “Biological Network” within the Human Cell Atlas [74], is recommended. We believe that this standardization roadmap can lead to the validation of organoid systems, and be optimized to maintain the quality of the research materials.

## 9. Clinical Application of Organoids

Organoid technology has the potential to be translated from the bench to bedside as a cell therapy and a cell transplantation, but several challenges must be addressed: (i) animal-derived sources in the derivation and culture of organoids need to be eliminated; (ii) mass expansion of organoids to clinically relevant numbers needs to be achieved; and (iii) immune rejection of organoids upon clinical transplantation needs to be prevented. Apart from these hurdles, organoid technology can lead to good current manufacturing practices to be recognized for clinical application [75].

## 10. Conclusions and Prospects

Many studies and preclinical trials have been performed to find effective biomarkers or for scoring the cancer system for diagnosis, monitoring of recurrence and metastasis, and treatment of each stage of GCs (Table 2). 

An initial trial was done with a drug targeting HER2; however, several clinical trials with drugs targeting HER2 failed to reach satisfactory outputs [83,84,85]. The use of drugs such as Trastuzumab, Pertuzumab, and Lapatinib did not generate reasonable results.

More recently, researchers and clinicians have incorporated data from TCGA and GEO to screen for target biomarkers and described a summary list of the possible candidate markers [7]. However, there remain several challenges in utilizing these markers in clinical applications. Here, we review the results of searching for the biomarkers for progression of GCs. Based on stemness reprogramming, the *TP53* mutation was examined [86], and stemness genes such as octamer-binding transcription factor *OCT4,* sex determining region Y-box transcription factor 2 (*SOX2*), and homeobox protein *NONOG* were addressed in GC cells or tissues [87]. Based on these three representative stemness factors, we next searched for stemness genes in GSCs in the antrum, fundus, and corpus regions (Lgr5, Axin2, Cxcr4, Cckbr, Bmi1, eR1, Mist1, Stmn1, Iqgap3, Tff2, and Cckbr; Figure 1 and Table 1 and Table 2) and examined the mutations in these basal and specific stemness-related genes. Thus, the combination of stemness genes and stomach stemness genes should be critical for searching for the relevant biomarkers. Next, we examined the driver mutation of the oncogenes and tumor suppressor genes, as well as the DNA mutation genes described in Section 5 and Table 1. In addition, alterations of microsatellites are critical for the induction of GC. The combined mutations and the altered expression of these marker combinations might become new biomarkers, but there are no reports so far identifying a link between these mutations. Nonetheless, this approach is required for identifying the roadmap of GC initiation and progression. A summary of the series of biomarkers possibly linked to the development process for a commitment to GC is presented (Figure 4).

Thus, cancer stemness genes and their mutations might assist the identification of novel biomarkers for GCs. The meta-analysis of OCT4, SOX2, NONG, and LGR5 and Bmi1 is summarized in Table 3. The association between CSC markers and the clinicopathological features reported in [10], indicated that OCT4, SOX2, and Bmi1 are significantly related. The data on Lgr5 and Nanog were not significant. The association between CSC markers and overall survival (OS) suggested that SOX2 is closely linked with worse OS in GC patients, whereas Bmil1, LGR5, and NANOG did not display any association. Similarly, an association between CSCs and disease-free survival (DFS)–relapse-free survival (RFS) was detected only for OCT4. The association between CSCs and cancer-specific survival (CSS)/disease-specific survival (DSS) was found for Bmi1 only. The available data for other genes were inadequate, and larger sample sizes may be required to find any significant connection.

At present, only few GCSC lines were available, including the cancer IPSCs clones from the Cell Engineering Division of RIKEN BioResource Research Center in Japan (Table 2). In addition, the gastric-cloned organoid lines were also reported [73,74].

As organoid and stem cell-induced pluripotent stem cell technologies continue to advance, biomarkers for specifying stem cells and cancer stem cells should be clearly identified to provide a precise language and consistent reproducibility, and to promote scientific progress within the field. At present, 3D-organoid technology is widespread within the scientific community; thus, the identification of well-defined, specific biomarkers will be important to promote standardization and validation of the technology. This, in turn, will help bring a consensus and promote the interactive and dynamic interchanges required to achieve the translational research to clinical applications.

## Figures and Tables

**Figure 1 jpm-12-00929-f001:**
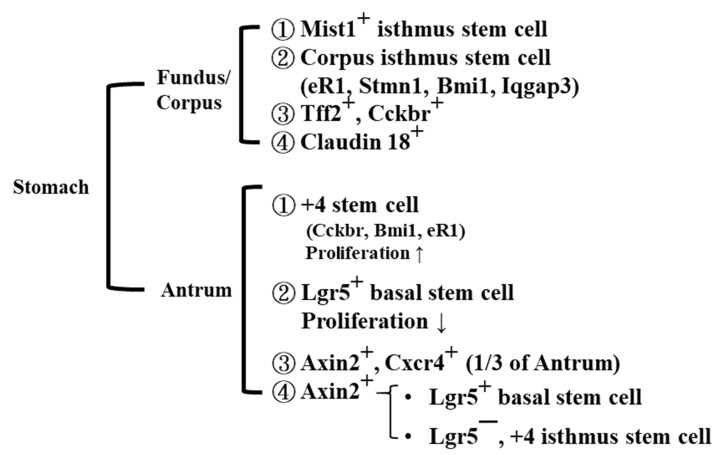
Representative stem cells are grouped in the stomach. Stem cells in both the antrum and fundus/corpus were assigned by the specific biomarkers as defining the stem cells, and these stem cells and progenitor cells can be mutations of the oncogene and the tumor suppressor genes to induce the neoplasia program.

**Figure 2 jpm-12-00929-f002:**
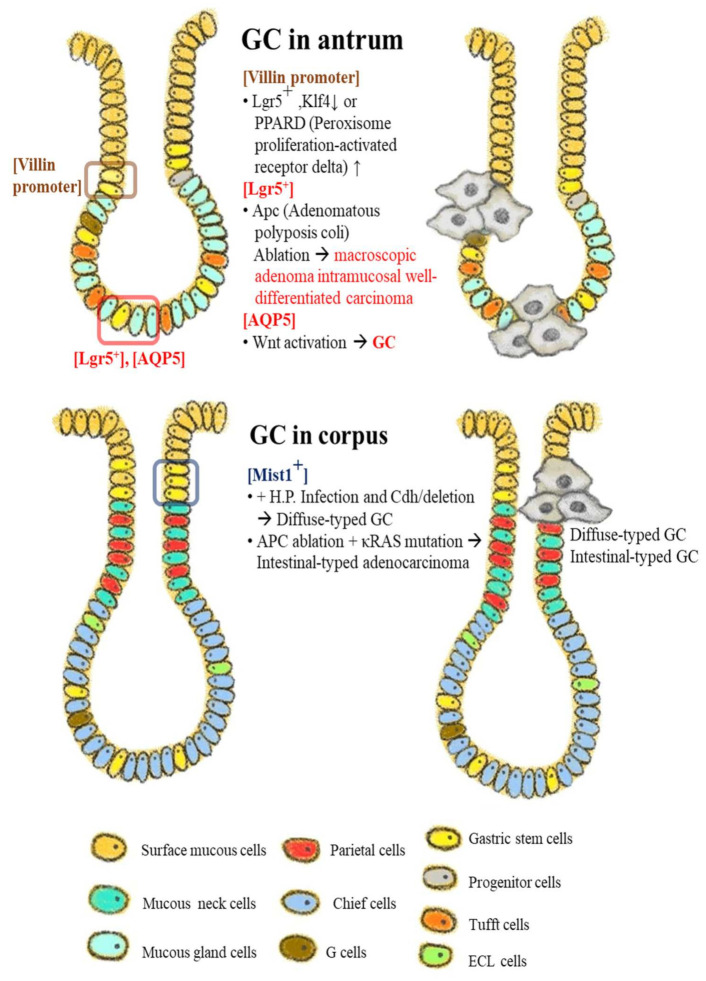
Schematic models of gastric stem cells and their specific markers in gastric carcinogenesis. The stem cell candidates or their origins specified by the respective markers were activated by the promoter activation, signaling activation, mutation, and ablation/deletion of the putative oncogenes to induce the gastric cancers in antrum and corpus regions. In antrum GC, light blue color indicates the mucous gland cells, and in corpus GC, medium blue-green color and blue color indicate the mucous neck cells and chief cells, respectively. Brown boxed area and red boxed area indicate the localized sites of villin promoter in the antrum region and the Lgr5/AQP active locus in the corpus region. This figure was modified from the original versions with permission from Hayakawa et al. [12], Stem cells and origins of cancer in the upper gastrointestinal tract; published by Cell stem cell, 2021 (5279230068983).

**Figure 3 jpm-12-00929-f003:**
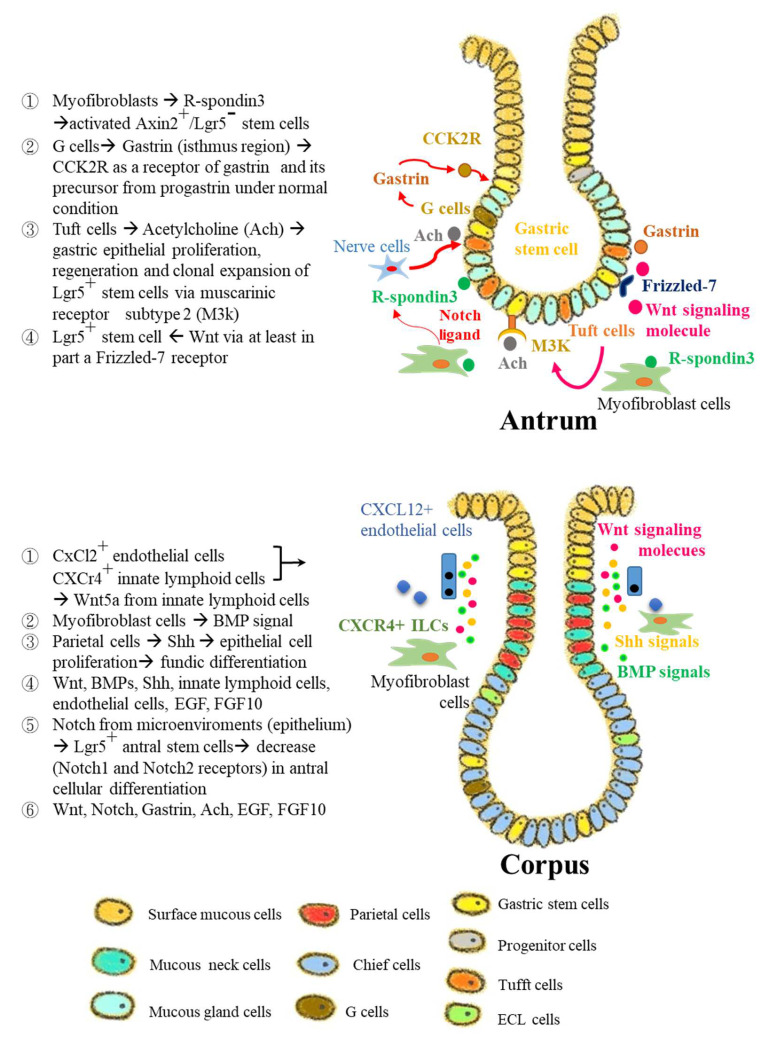
Interaction between gastric stem cells and their microenvironmental factors. In the antrum, Notch signal affects the proliferation and differentiation of stem cells and differentiation. Acetylcholine (Ach) and gastrin also contributed to niches factors nearby. Light blue indicates mucous gland cells. In the corpus region, Shh and BMPs signaling are localized at isthmus and the neck of fundus. Stromal Cxcl12^+^ endothelial cells and Cxcr4^+^ innate lymphoid cells, also contributed to the niches of the corpus stem cells in part via Wnt5a signal. Other factors such as Egf, Fgf10, and Wnt play a critical role in the control of self-renewal and differentiation of GSCs in the basement glands of the antrum and the isthmus regions of corpus. Light green and blue indicate mucous neck cells and chief cells, respectively. This figure was modified from the original versions with permission from Hayakawa et al. [12], Stem cells and origins of cancer in the upper gastrointestinal tract; published by Cell stem cell, 2021 (5279230068983).

**Figure 4 jpm-12-00929-f004:**
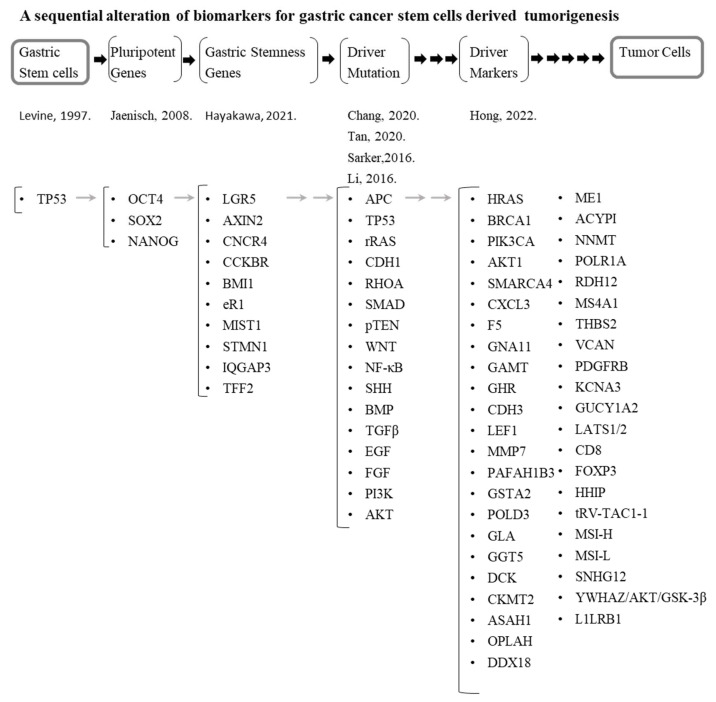
A possible road map to seek for the biomarkers for developing of human gastric cancer. The lineage of alterations of gastric stem cells are summarized for each step of mutations and epigenetic alterations of cancer-competent genes. Gastric stem cells cited as [87]; Pluripotent genes are cited as [87]; Driver mutations are cited as [32,41,60,61]; Driver markers have been cited recently in [7]. ACYP1: Acylphosphatase 1; ASAH1: N-Acylsphingosine amidohydrolase 1; BRCA1: Breast cancer type 1 susceptibility protein 1; CDH1: Cadherin 1; CDH3: Cadherin 3; CXCL3: C-X-C motif chemokine ligand 3; CKMT2: Creatine kinase, mitochondrial 2; DCK: Deoxycytidine kinase; DDX18: DEAD-box helicase 18; F5: Coagulation factor V; GAMT: Guanidinoacetate N-methyltransferase; FOXP3: Forkhead box P3; GGT5: Gamma-glutamyltransferase 5; GHR: Growth hormone receptor; GLA: Galactosidase alpha; GNA11: G-protein subunit alpha 11; GSTA2: Glutathione S-transferase alpha 2; GUCY1A2; Guanylate cyclase 1 soluble subunit alpha 2; HHIP: Hedgehog interacting protein; KCNA3: Potassium voltage-gated channel subfamily A member 3; LATS1/2: Large tumor suppressor kinase 1/2; LEF1: Lymphoid enhancer binding factor 1; MMP7: Matrix metallopeptidase 7; OPLAH: 5-Oxoprolinase, ATP-hydrolyzing; LILRB1: Leukocyte immunoglobulin-like receptor B1; ME1: Malic enzyme 1; MS4A1: Membrane spanning 4-domains A1; MSI-H: Microsatellite instability-high; MSI-L: Microsatellite instability-low; NNMT: Nicotinamide N-methyltransferase; PAFAH1B3: Platelet activating factor acetylhydrolase 1b catalytic subunit 3; PDGFR3: Platelet-derived growth factor receptor beta; POLD3: DNA polymerase delta 3, accessory subunit; POLR1A: RNA polymerase I subunit A; RDH12: Retinol dehydrogenase 12; SMARCA4: SWI/SNF-related, matrix-associated, actin-dependent regulator of chromatin, subfamily a, member 4; SNHG12: Small nucleolar RNA host gene 12; THBS2: thrombospondin 2; VCAN: Versican; tRV-TAC1-1: tRNA-Val ((anticodon TAC) 1-1); and YWHAZ: Tyrosine 3-monooxygenase/tryptophan 5-monooxygenase activation protein zeta.

**Table 1 jpm-12-00929-t001:** Gastric cancer stem cells and iPS derived gastric cancer stem cells.

Region	Stem Cell	Driver Mutations/Cancers
**[Antrum]**	**1.** **LGR5 basal stem cells** **2.** **+4 stem cells** **3.** **Tumor-resident LGR5** **(CLAUDIN18)**	• **SMAD, PTEN →** **Invasive cancer** **↑****κRAS/TP53** • **APC →** **Adenoma Intramucosal dysplasia** • **WNT → Invasive cancer, Advance GC** **RAS** **TP53**
**[Fundus/corpus]**	**1.** **MIST1 isthmus stem cells**	• **κRAS → Metaplasia → APC → intestinal-type cancer** • **CDH1 → + inflammation → Intramucosal signet-ring cancer → RHOA, TP53 → Invasive diffuse-type cancer**

**Table 2 jpm-12-00929-t002:** Driver mutation of marker genes of CSCs in development of gastric cancer.

References	Original Cells	Methodology	Teratoma Tumor Formation	Features
Miyoshi et al. [76] Proc Natl Acad Sci USA 107(1): 40–45. (2010)	Human gastrointestinal cells	Lentivirus Retrovirus + Lipofectamine OSLM, Nanog, LIN 28, BCL2, κRAS, and tumor suppressor shRNA	Yes	To 5-Fu and differentiation inducing resistance
Hoshino et al. [77] Int J Oncol 40(5):1423–30. (2012)	HCT116 or HCT116 (TP53^−/−^)	Lentivirus OSLN+ hypoxia	Yes	Highly aggressive
Oshima et al. [78] PLoS ONE 9(7): e101735. (2014)	Colorectal cancer SW480 DLD-1	Retrovirus OSK	Yes	CSCs features enhanced tumorgenicity
Zhu et al. [79] Int J Med Sci 17(13):2040–2051. (2020)	Human gastric cancer cell line SGC7901 Xue et al. [80] (J Cell Biochem 113:302-312, 2012)	Selection of SOX2, OCT4, CD44, CD133	Yes	CSC-G clone
RIKEN BRC Cell Engineering Division [81] https://cell.brc.riken.jp/en/ (accessed on 13 January 2022)	Gastric cancer patents	iPS methodology	Yes	HPS3574~3585 HPS3558~3573

CSS: Cancer-specific survival; DFS: Disease-free survival; DSS: Disease-specific survival; NR: Not reported; TNM stage: Cancer stages based on the NCI, USA; NOS score is based by the Newcastle-Ottawa Scale (range 1–9) [82]; OS; Overall survival.

**Table 3 jpm-12-00929-t003:** Meta-analysis of gastric CSC markers [10,88,89].

CSC Markers	Country Year	Patients Age (Year)	Sample Size (Number)	TMM Stage	Outcome	NOS * Score	Reference
1. OCT4	China ~2010	62	412	I–IV	OS	7	[90]
2. OCT4	Iran ~2014	59.3	40	I–IV	OS	6	[91]
3. OCT4	Egypt ~2016	44	45	I–III	OS, DFS	6	[92]
4. OCT4	China ~2014	NR	158	I–IV	OS	8	[93]
5. OCT4	Japan ~2012	NR	290	I–IV	OS	8	[94]
6. OCT4	China, Korea, Slovenia, Iran Denmark, Japan ~2020	NR	5198	I–IV	OS	6-8	[88]
7. SOX2 OCT4 NANOG	Japan ~2012	NR	290	I–IV	OS	6	[94]
8. SOX2 OCT4 NANOG	China ~2009	55	69	I–III	OS, DFS	8	[95]
9. SOX2 ALDH	China ~2013	63	122	I–IV	OS	8	[96]
10. SOX2	China ~2004	58	50	I–IV	OS	7	[97]
11. SOX2	Portugal ~2010	66.5	201	I–IV	OS	8	[98]
12. LGR5	Germany ~2009	68	487	I–IV	OS	6	[99]
13. LGR5	China ~2009	61	257	I–IV	OS	8	[100]
14. LGR5	China ~2004	60	318	I–IV	OS	7	[101]
15. LGR5	Korea ~2006	60	456	I–IV	CSS	6	[102]
16. LGR5	China ~2014	60	100	I–III	OS	8	[89,103]
17. BMI1	China ~2002	60	146	I–IV	OS	7	[104]
18. BMI1	China ~2004	60	75	I–IV	OS	7	[105]
19. BMI1	China ~2005	60	219	I–IV	DSS	7	[106]
20. BMI1	China NR	60	352	I–IV	OS	6	[107]
21. BMI1	China ~2001	55	309	NR	OS	7	[108]
22.NANOG	China NR	50	105	I–IV	OS	7	[109]

* NOS = Newcastle-Ottawa Scale.

## Data Availability

Please contact the corresponding author for such requests.

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
