# Peer review of "Stem Cell Biomarkers and Tumorigenesis in Gastric Cancer"

_jpm, 2022, doi:10.3390/jpm12060929_

Round 1

Reviewer 1 Report

The review article entitled: "Stem Cell Biomarkers and Tumorigenesis in Gastric Cancer" provides a thorough description of gastric stem cells. However, has no clear line of discussion, any new perspectives or logics to apply to the field, nor does it add to the previous existing literature regarding these GSCs. 

Author Response

I am so sorry not to make you comfortable. However, I am not sure what points which the reviewer  criticized us. The decision for withdraw is too simple and not so kind to the authors. The story of this text is simple to show the mutation of the possible stem cell markers and the driver mutation to initiate the gastric cancer development. This combination and the mutations of other niches mutations might be the possible candidates for the new cancer markers. I think this is a new perspective and a logic. I also cited the gastric cancer stem cell markers from normal stem cells which are possible mutated and from the cancer patient-derived iPSCs. Thus, I did not cite the references generated by the stomach tissues or the Biobank because of the problems of heterogeneity and no definitions of the stem cells as the sources. We rewrite and add the new information. Please see them.

Reviewer 2 Report

In this manuscript, the authors provide a comprehensive review of gastric cancer stem cell biomarkers and elucidate their significance. It also discussed the 3D culture of organoids and its potential application in the treatment of gastric cancer. The work was informative and provided comprehensive insights. However, there are several concerns that need to be addressed by the authors, as summarized below.

Major concerns:

  1. What is the difference between gastric cancer stem cells and stem cells from other tumor types? Do the biomarkers listed in the text have similar functions in other cancer types? In the TCGA cohort, what is the prognosis of gastric cancer patients with high expression of these biomarkers? Please expand the description.
  2. Are there any gastric cancer stem cell lines currently available? What is its source and which scientist isolated it? Please list the table to make a statement.

Minor comment:

  1. Pay attention to details in your writing, such as punctuation. Please check the paragraph numbers in sections 2, 3, 4, and 5.
  2. There are still some English grammar problems in the paper, which need to be modified to improve the readability of the article.
  3. Conjunctions should be used in their correct form. e.g.: in line 375 of page 12, the phrase “stem cell/introduced” should be written as “stem cell-introduced”.
  4. Please keep the font size and format in the figures consistent.

Reviewer 3 Report

Cancer biomarkers are currently predominant in most research groups. This is fundamentally because of cancer infections consequences in human existence. Consequently, this information is imperative in further developing innovation in diagnostics, subsequently work on the adequacy of the medicines.

The manuscript presents some recent studies upon biomarkers associated with gastric cancer, mainly being focused on tracing models for lineage analysis of GC stem cells. This paper encompasses a novel topic that will be of interest to the researchers. I think this paper is potentially suitable for publication in Journal of Personalized Medicine after considering a major revision. I have some queries and suggestions for improving the overall presentation.

1) The introduction is missing background on these biomarkers and their levels. Most of the introduction is written as very general and not focused on these analytes. It would be valuable for the article to have some general information about the biomarkers.

2) Please check the spelling of the text. There are some typing errors.  Also, please check the paragraph spacing in text.

3) Figure 2 and 3. There is a different shade of blue in the figure, but without mention about it in the legend. Is it an error or the authors forgot to mention it in the legend?

4) It would be beneficial for readers, if the authors could make a table of all abbreviations.  

5) Line 144. There is too little information about this topic. The authors should provide more details, explanations about how Mist1-CreERT2 mice were used to identify corpus stem cells. Also, the details should be sustained by literature references.

6) Some clinical studies or study references that complement the paragraphs about the biomarkers would be welcomed.

Author Response

Editor-in-Chief

Journal of Personalized Medicine

Editorial office

MDPT, St. Alban-Anlage 88, 4052 Basel

Switzerland

Journal: Journal of Personalized Medicine
Special Issue: Biomarker Identification and Application of Cancer Stem Cells

Manuscript ID: jpm-1643654R1

May 20th, 2022

Dear Editor:

We are pleased to resubmit the review article by Dr. Kenly Wuputra entitled “Stem cell biomarkers and tumorigenesis in gastric cancer” for publication to the special issue “Biomarker identification and application of cancer stem cells”.

          As suggested by the reviewers, we have incorporated all criticism raised by reviewers and add the additional Table 3 and Figure 4. This review shows the possible biomarker for the gastric cancer stem cells which included the makers of stem cells, driver mutations, possible reported markers (Table 1 and Figure 4). There are no reports to show the combination of the cancer stem cells and stemness related genes and their driver mutations to become the new biomarker targets for gastric cancer. The studies of these normal stem cells markers and the cancer stem cells markers have not reported in depth (Table 3). Thus, in a future we need to concentrate on these target genes to define the stem cell plasticity as the novel biomarkers. I hope these revisions will be satisfactory to the reviewers.

Each point raised by the reviewer are answered as following.

「Reviewer 」

Cancer biomarkers are currently predominant in most research groups. This is fundamentally because of cancer infections consequences in human existence. Consequently, this information is imperative in further developing innovation in diagnostics, subsequently work on the adequacy of the medicines.

The manuscript presents some recent studies upon biomarkers associated with gastric cancer, mainly being focused on tracing models for lineage analysis of GC stem cells. This paper encompasses a novel topic that will be of interest to the researchers. I think this paper is potentially suitable for publication in Journal of Personalized Medicine after considering a major revision. I have some queries and suggestions for improving the overall presentation.

  • The introduction is missing background on these biomarkers and their levels. Most of the introduction is written as very general and not focused on these analytes. It would be valuable for the article to have some general information about the biomarkers.

(Answer)

As suggested by the reviewer, we added to write the general information about the biomarkers in the introduction section.

  • Please check the spelling of the text. There are some typing errors.  Also, please check the paragraph spacing in text.

(Answer)

As suggested by the reviewer, we corrected the typing errors and paragraph spacing in the text.

  • Figure 2 and 3. There is a different shade of blue in the figure, but without mention about it in the legend. Is it an error or the authors forgot to mention it in the legend?

(Answer)

As suggested by the reviewer, we have added the explanation of the different shaded blue marks.

  • It would be beneficial for readers, if the authors could make a table of all abbreviations.

(Answer)

 As suggested by the reviewer, we have added Abbreviation section in the text.

  • Line 144. There is too little information about this topic. The authors should provide more details, explanations about how Mist1-CreERT2 mice were used to identify corpus stem cells. Also, the details should be sustained by literature references.

(Answer)

As suggested by the reviewer, we have added the explanation section with references.

  • Some clinical studies or study references that complement the paragraphs about the biomarkers would be welcomed.

(Answer)

As suggested by the reviewer, we have added the database information of the biomarkers which were examined. The clinical studies were not performed yet. Thus, here we demonstrated the data-base derived summary (Table 3). 

Round 2

Reviewer 1 Report

In the previous round of review, major revisions was requested to present a clear line of discussion, new perspectives or logics to apply to the field. So it would significantly add to the previous existing literature regarding these GSCs.  However, no such major revision is provided.

Author Response

Editor-in-Chief

Journal of Personalized Medicine

Editorial office

MDPT, St. Alban-Anlage 88, 4052 Basel

Switzerland

Journal: Journal of Personalized Medicine
Special Issue: Biomarker Identification and Application of Cancer Stem Cells

Manuscript ID: jpm-1643654R1

May 20th, 2022

Dear Editor:

We are pleased to resubmit the review article by Dr. Kenly Wuputra entitled “Stem cell biomarkers and tumorigenesis in gastric cancer” for publication to the special issue “Biomarker identification and application of cancer stem cells”.

          As suggested by the reviewers, we have incorporated all criticism raised by reviewers and add the additional Table 3 and Figure 4. This review shows the possible biomarker for the gastric cancer stem cells which included the makers of stem cells, driver mutations, possible reported markers (Table 1 and Figure 4). There are no reports to show the combination of the cancer stem cells and stemness related genes and their driver mutations to become the new biomarker targets for gastric cancer. The studies of these normal stem cells markers and the cancer stem cells markers have not reported in depth (Table 3). Thus, in a future we need to concentrate on these target genes to define the stem cell plasticity as the novel biomarkers. I hope these revisions will be satisfactory to the reviewers.

Each point raised by the reviewer are answered as following.

[Reviewer 1]

In the previous round of review, major revisions was requested to present a clear line of discussion, new perspectives or logics to apply to the field. So it would significantly add to the previous existing literature regarding these GSCs. However, no such major revision is provided.

(Answer)

Thank you so much for your kind instruction how to revise my review article. The logic and the new perspectives for the Biomarkers for human gastric cancer were summarized in the new Figure 4 and the text in the conclusion and perspective section. In this article we focus upon the biomarkers of gastric cancer stem cells and its stem cells niches as well as the driver mutations. Thus, the papers avoided the papers which described the gastric tentative stem cells which were identified by the biomarker genes generated from the bulk tissues. Because of its heterogeneity of the gastric cancer cell types, we avoided these papers. We only concentrated the stem cells derived cancer stem cells and stem cells niches to identify the cell fates for the cancer induction. These stem cell markers are altered by the driver mutations including the epigenesis and the genome mutations. The possible database-oriented biomarker genes were also summarized based on the normal pluripotent stemness genes (see Table 3). Moreover, the gastric cancer was generated the heterogeneity of the distinct parts of the stomach which are divided by the specific stemness genes as the markers (Table 1). This combination was focused upon the stemness related biomarkers and not included the unknown origin of the CSCs. This strategy is illustrated In Figure 4 and Table 1. We also discussed about his perspectives in discussion section.

Reviewer 2 Report

Modifications have been completed. This paper can be published.

Author Response

Editor-in-Chief

Journal of Personalized Medicine

Editorial office

MDPT, St. Alban-Anlage 88, 4052 Basel

Switzerland

Journal: Journal of Personalized Medicine
Special Issue: Biomarker Identification and Application of Cancer Stem Cells

Manuscript ID: jpm-1643654R1

May 20th, 2022

Dear Editor:

We are pleased to resubmit the review article by Dr. Kenly Wuputra entitled “Stem cell biomarkers and tumorigenesis in gastric cancer” for publication to the special issue “Biomarker identification and application of cancer stem cells”.

          As suggested by the reviewers, we have incorporated all criticism raised by reviewers and add the additional Table 3 and Figure 4. This review shows the possible biomarker for the gastric cancer stem cells which included the makers of stem cells, driver mutations, possible reported markers (Table 1 and Figure 4). There are no reports to show the combination of the cancer stem cells and stemness related genes and their driver mutations to become the new biomarker targets for gastric cancer. The studies of these normal stem cells markers and the cancer stem cells markers have not reported in depth (Table 3). Thus, in a future we need to concentrate on these target genes to define the stem cell plasticity as the novel biomarkers. I hope these revisions will be satisfactory to the reviewers.

Each point raised by the reviewer are answered as following.

[Reviewer ]

Modifications have been completed. This paper can be published.

(Answer)

Thank you so much for your decision.

Reviewer 3 Report

The manuscript was improved and some terms were better explained. But, there are still some minor issues to be resolved before it can be published in the journal. I also advise the authors to benefit from grammatical editing from a native English speaker.

Mostly, despite English corrections, there are 2 paragraphs from pages 2 and 4 that must be improved or add more details.  On page 2, line 59, please specify several types of diseases, due to the fact that you mentioned in the manuscript the following " The initial hits of SPEM and/or 57 IM used to be accompanied with the accumulated gene mutations in chronic 58 gastritis, and then derived to the diseases to adenocarcinoma severely@.  On page 4, subtitle 2.4, lines 181-183, please provide more details about "CxcR4+" biomarker, in order to improve the understanding of this particular gastric cancer biomarker, and to help readers better understand. 

Author Response

Editor-in-Chief

Journal of Personalized Medicine

Editorial office

MDPT, St. Alban-Anlage 88, 4052 Basel

Switzerland

Journal: Journal of Personalized Medicine
Special Issue: Biomarker Identification and Application of Cancer Stem Cells

Manuscript ID: jpm-1643654R1

May 26th, 2022

Dear Editor:

We are pleased to resubmit the review article by Dr. Kenly Wuputra entitled “Stem cell biomarkers and tumorigenesis in gastric cancer” for publication to the special issue “Biomarker identification and application of cancer stem cells”. As suggested by the reviewer, we have corrected our manuscript. We also asked to edit this manuscript by the company so that the native English speaker revised our manuscript.

The precise reply to the reviewer 3 is follow.

[Reviewer 3]

The manuscript was improved and some terms were better explained. But, there are still some minor issues to be resolved before it can be published in the journal. I also advise the authors to benefit from grammatical editing from a native English speaker.

Mostly, despite English corrections, there are 2 paragraphs from pages 2 and 4 that must be improved or add more details.  On page 2, line 59, please specify several types of diseases, due to the fact that you mentioned in the manuscript the following " The initial hits of SPEM and/or 57 IM used to be accompanied with the accumulated gene mutations in chronic 58 gastritis, and then derived to the diseases to adenocarcinoma severely@. 

(Reply)

Thank you for kind reminding. We added the underlining text in to paragraph 2 to provide more detail related with the diseases.

The initial hits of SPEM and/or IM used to be accompanied with the accumulated gene mutations in chronic gastritis, and then derived to the diseases to adenocarcinoma severely [2,4]. To specify the diseases caused by IM, mucosal atrophy characterized by losing glandular elements resulted in replacing the connective tissue (fibrosis), or metaplastic cells (neoplastic lesion). These neoplastic lesions injured the gastric mucosa correlated with H. pylori (infection) infection, resulted in chronic inflammation (chronic gastritis). Precancerous lesions also let to SPEM, strongly correlated with GC progression [2]. The Cancer Genome Atlas (TCGA) has presented to help the understanding of gene mutation pattern in GC. ---

On page 4, subtitle 2.4, lines 181-183, please provide more details about "CxcR4+" biomarker, in order to improve the understanding of this particular gastric cancer biomarker, and to help readers better understand. 

(Reply)

We are sorry that we made a typing-error in this part. This is Cckbr; stem cells, not CxcR4+ cells. Thus, we added the following explanation in the revised test.

Cckbr+ stem cells

One type of +4 stem cells, Cckbr+ stem cells, committed to the asymmetric division to generate the progeny with canonical notch ligand 1 either Numb or Delta for long term proliferation and division [32]. By contrast, in the cases of inflammation and carcinogenesis, these Cckbr+ cells can be converted to the symmetric cell division to expand for tissue regeneration, probably by the loss of gastrin-dependent signals [32]. Gastrin secreted from G cells resided near Cckbr+ stem cells which expressed gastrin receptor. In the case of the loss of gastrin, Cckbr+ stem cells became more proliferate and started to divide symmetrically, indicating that gastrin kept maintaining quiescence or inhibited antrum stem cells.  

The contents of this manuscript are not now under consideration for publication elsewhere. All authors have directly participated in the planning, execution, or data analysis of the study. All authors of this paper have read and approved the final version submitted and co-authors of this paper have read and approved the final version submitted and co-authoring in this manuscript.

Thank you very much for your consideration of our manuscript.

Sincerely yours,

Kazunari K(Kazushige). Yokoyama, Ph.D.

Graduate Institute of Medicine

Kaohsiung Medical University

100 Shih-Chuan 1st Road, San Ming District

807 Kaohsiung, Taiwan

Phone:+886-7-312-1101, ext. 2729

Fax:+886-7-313-3849

                                                        http://kazu.dlearn.kmu.edu.tw/